# Uterine Transcriptome: Understanding Physiology and Disease Processes

**DOI:** 10.3390/biology12040634

**Published:** 2023-04-21

**Authors:** Gregory W. Kirschen, Kamran Hessami, Abdelrahman AlAshqar, Sadia Afrin, Bethlehem Lulseged, Mostafa Borahay

**Affiliations:** 1Department of Gynecology & Obstetrics, Johns Hopkins University, Baltimore, MD 21287, USA; 2Maternal Fetal Care Center, Boston Children’s Hospital, Harvard Medical School, Boston, MA 02115, USA; 3Department of Obstetrics, Gynecology and Reproductive Sciences, Yale School of Medicine, New Haven, CT 06510, USA; 4School of Medicine, Johns Hopkins University, Baltimore, MD 21205, USA

**Keywords:** transcriptomics, leiomyomas, fibroids, endometriosis, mRNA, microRNA, RNA sequencing

## Abstract

**Simple Summary:**

RNA sequencing of the human endometrium and myometrium during normal menstruation and in gynecological disorders including endometriosis, adenomyosis, fibroids, and recurrent pregnancy loss has proved instrumental in advancing our understanding of the cellular complexity and dynamics that underlie these processes. Through the use of single-cell RNA sequencing, the field has uncovered multiple novel endometrial cell subpopulations that include immune cells, epithelial cells, and stem/progenitor cells that work in synchrony to coordinate the menstrual cycle and blastocyst implantation, and whose aberrant activity likely helps explain the aforementioned pathological states, which may aid in development of novel therapeutic interventions. In this review, we summarize recent advances in RNA biology pertaining to understanding the uterus through its transcriptome.

**Abstract:**

In recent years, transcriptomics has enabled us to gain a deeper understanding of fundamental reproductive physiology, including the menstrual cycle, through a more precise molecular analysis. The endometrial mRNA transcript levels fluctuate during the normal menstrual cycle, indicating changes in the relative recruitment and abundance of inflammatory cells, as well as changes in the receptivity and remodeling of the endometrium. In addition to providing a more comprehensive understanding of the molecular underpinnings of pathological gynecological conditions such as endometriosis, leiomyomas, and adenomyosis through RNA sequencing, this has allowed researchers to create transcriptome profiles during both normal menstrual cycles and pathological gynecological conditions. Such insights could potentially lead to more targeted and personalized therapies for benign gynecological conditions. Here, we provide an overview of recent advances in transcriptome analysis of normal and pathological endometrium.

## 1. Background

The human endometrium is a dynamic tissue that undergoes repeated growth and regression with each menstrual cycle [1,2]. Structurally, the endometrium consists of a rich supply of microvasculature derived from myometrial arcuate arteries dividing into feeding radial arteries that give rise to spiral arterioles supplying the functional layer of the endometrium [3]. Underlying this functional layer is the basal layer, where endometrial glands are surrounded by dense stroma, and where adult stem/progenitor cells lie, contributing to endometrial regeneration [4]. Estrogen and progesterone act through nuclear receptors to effect changes on gene expression in the endometrium, most notably including estrogen-mediated mucosal proliferation during the proliferative phase of the menstrual cycle, and progesterone-mediated, prostaglandin E2 (PGE2)-dependent decidualization of endometrial stroma, preparing the endometrium for implantation during the secretory phase of the menstrual cycle [5,6].

Menstrual blood is a valuable source for obtaining mesenchymal/stromal stem cells which have a strong potential for self-renewal, high rates of proliferation, pluripotency, migratory and immunomodulatory activities under inflammatory, tumor, and tissue-injury conditions [7,8]. Due to the diverse clinical applications, menstrual mesenchymal cells have gained prominence since their discovery in the past two decades in different gynecological diseases.

Simultaneously discovering these progenitor cells, several approaches such as genomics, epigenomics, transcriptomics, and proteomics have been applied to better understand physiologic and pathologic conditions associated with human endometrium. The new ‘–omics’ technology has emphasized understanding and characterizing ways to identify the transcriptomic physiological profile in each menstrual cycle [9], assessing endometrial profiles of fertile patients, i.e., endometrial receptivity window [10], and those with infertility or recurrent implantation failure [11,12]; and finally, comparisons of profiles between healthy women and women with endometrial pathologies such as endometriosis, adenomyosis, and endometrial cancer [13,14,15]. 

For this article, we performed an extensive literature review using the PubMed MEDLINE database using relevant search terms that included “transcriptome”, “transcriptomics”, “RNA”, “RNA sequencing” “microRNA”, “menstrual cycle”, “endometriosis”, “leiomyomas”, “fibroids”, “adenomyosis”, “recurrent implantation failure”, and “gynecological”. Additional relevant articles have been added during the revision process according to expert reviewer feedback.

## 2. Microarrays and RNA-Sequencing

Microarray-based gene expression technology which allows parallel evaluation of the expression of numerous genes has been one of the most commonly used tools for transcriptome analysis. Endometrial transcriptomics has been applied to many aspects of endometrial physiology and pathophysiology, including the normal menstrual cycle [16,17]. While any given study provides numerous candidate genes to explore, the number of genes, which have been identified in more than one study as potential biomarkers in endometrial physiology and pathophysiology, has remained somewhat limited. This can be attributed to the variation in experimental design, timing and procedures of endometrial sampling, selection criteria of patient and control groups, and strategies used for data processing [18]. 

Most transcriptome analyses have utilized biopsies of entire endometrial tissue containing different cell types. Therefore, the measured mRNA abundance represents an average of all cell types present. In addition to microarrays, there is the emerging alternative of RNA-seq, in which all RNAs are sequenced and most of the genes being expressed can be revealed. RNA-seq technology identifies more exons and alternative splicing events than microarray because it is entirely independent of prior knowledge. Microarrays usually fail to pick up an average of 25% of low-expression genes, but such low-abundance transcripts are detected in RNA-seq reads [19]. Previous studies noted that single-cell RNA-seq has other utilities such as (i) detecting cell-to-cell variability and mapping possible sub-populations [20], (ii) discovering possible rare cell types [21], and (iii) studying clinically relevant but rare endometrial adult stem cells [22]. Single-cell RNA-seq is considered a novel and promising approach to endometrial expression studies with an enormous potential for discoveries and an improved understanding of basic mechanisms governing tissue function. 

## 3. Endometrial Transcriptome and the Normal Menstrual Cycle

### 3.1. Menstruation

The onset of the menstrual period corresponds to the first day of the cycle, usually lasting for 3–5 days. This phenomenon is associated with the separation of the functional layer of the endometrium due to a sharp drop in the levels of sex hormones [23]. 

The menstrual period is a physiological state in which significant changes occur at the transcriptomic level. The principal processes involved are tissue interruption and apoptosis. Analysis of temporal gene expression patterns has shown a peak in the expression of the genes associated with apoptosis, inflammation, signal transduction, transcription, and DNA repair [24]. These comprise natural cytotoxicity-triggering receptor 3 (NCR3) [25], Wnt family members (Wnt5a and Wnt7a) [25] and the family of matrix metalloproteinases (MMPs) [24]. 

NCR3, first reported by Ponnampalam et al. [9], is increased towards the late-secretory phase and was sustained in the menstrual phase. Subsequently, its level decreased in the proliferative phase. This gene also plays a prominent role in the inflammatory response in the recognition of no-HLA ligands by natural killer (NK) cells [26]. NCR3 is activated by the presence of NK cells and mimics the expression profile of NK cells in the endometrium [9]. Wnt5a and Wnt7a are two significant genes associated with that stage and belong to the Wnt pathway; their increased expression level during the menstrual phase diminishes in the proliferative phase, probably due to an increase in the expression of specific Wnt inhibitors. Furthermore, it has been revealed that these genes are also most important in uterine glandular development [27].

*MMP* gene expression begins to surge toward the late secretory phase before falling back to low levels in the late proliferative phase [25]. MMPs are mainly involved in tissue desquamation [28]. This profile has been observed for *MMP1, -3* and *-10*, which are up-regulated in the menstrual compared to the proliferative phase. On the contrary, endometase (*MMP26*) has an increased expression in the proliferative phase, probably due to the tissue remodeling occurring during this phase [25]. 

Other genes identified in the premenstrual phase with high expression levels are protease-activated receptor type 1 (*F2R* (*PAR-1*)) and lysyl oxidase (*LOX*) [24]. The thrombin receptor (PAR-1) is responsible for facilitating many cardiovascular system functions, and it is positioned on the surface of endothelial cells and platelets. PAR-1 activation can result in platelet aggregation, vascular smooth muscle mitogenesis and vasoconstriction. It was also revealed that PAR-1 plays an important role in the events preceding and is, at least to some extent, accountable for menstrual bleeding [24]. On the other hand, the *LOX* gene encodes an amino oxidase, which is vital for collagen deposition procedure and many wound-healing-related functions; thus, it is probable that it plays a prominent role in endometrial repair [24]. 

### 3.2. Proliferative and Secretory Phases 

Microarray analysis has been used to study several genes, including *CCL18*, *MT2A*, *F2RL2*, *PLIN2*, and *TGFB2*, whose expressions are increased during the early proliferative phase. These genes are needed to regenerate the functional endometrial layer after menstruation [29]. Expression of genes involved in cell renewal processes such as cell proliferation, cell survival, and regulation of differentiation (such as *IHH*, *SERP4*, *PGR*, *SNRD14E*, and *GSTM1*) is increased during the mid-proliferative phase [29]. During the late proliferative phase, the upregulation of *AGTR2*, *HMGIC*, *C9orf131*, *SNORA23*, and *CRIM1* genes is linked with inhibition of the cell growth, extracellular matrix remodeling, and cellular differentiation [30]. However, Petracco et al. discovered low expression levels of genes that are associated with NK cell function, such as *KIR2DL3* and *KLRC3*, in the late proliferative phase, highlighting a diminished immune response in phase of the cycle, which happens to favor embryo implantation [29]. Furthermore, genes involved in tissue remodeling (*MMP26* and *TFF3*), cell differentiation (*HOXA10* and *HOXA11*), vasculogenesis (*GJB6, HOXB7*, and *sFRP*), and angiogenesis (*CXCR4, CDH5, ENG*, and *PECAM1*) are increasingly expressed during the proliferative phase [25]. Figure 1A displays representative gene cluster profiles (non-exhaustive list) across the menstrual cycle. Wang et al. provided a transcriptomic atlas of the human endometrium during the menstrual cycle, which scientists can use to ask a myriad of questions related to menstrual physiology (see NCBI’s Gene Expression Omnibus series accession code GSE111976) [31].

Next-generation sequencing has also studied the profile of the gene expression of endometrial tissue during the secretory phase [32]. Regarding the particular changes in the transcriptional profiles of each cell type residing in the endometrium, it has been shown that there is an abrupt change in the transcriptomic profile of the unciliated epithelia in the secretory phase (increased expression of *PAEP*, *GPX3*, and *CXCL14*). In contrast, stromal cells display a more continuous phase transition (increased expression of *DKK1* and *CRYAB*). 

### 3.3. Single-Cell Transcriptomics throughout the Menstrual Cycle

Advancements in single-cell transcriptomics have allowed us to characterize unique changes in individual endometrial cell types that work in synchrony to regulate the menstrual cycle. In effort to characterize the normal menstrual cycle in a group of 19 healthy ovum donors, Wang et al. [31] analyzed transcriptome profiles from endometrial biopsies with single-cell resolution. These investigators grouped cells into categories based on canonical markers: stromal fibroblasts, endothelial cells, macrophage, and lymphocyte. Beyond this, they identified two subtypes of epithelial cells, including ciliated and non-ciliated cell populations. For instance, while stromal fibroblasts upregulate genes such as *FOXO1* and *IL15* in promotion of decidualization, luminal epithelial cells alter their expression of proliferation-associated genes from the proliferative phase to the secretory phase. Each cell type displays a unique and typical pattern of transcriptome-wide gene expression that fluctuates during the menstrual cycle in concert to promote normal endometrial physiology. Moreover, Wang et al. [31] determined based on cell markers that vascular smooth muscle cells display characteristic stem/progenitor cell features. Overall, this study showed that single-cell transcriptomics is a useful tool for identifying important intracellular signaling pathways activated with temporal and cell type-specific resolution. This technology can also be used to discover novel cell types based on divergent transcriptome signatures among neighboring cells within a given tissue. 

Complementing these data, Garcia-Alonso et al. [33] performed single-cell RNA sequencing (scRNA-seq) and single-nucleus RNA sequencing (snRNA-seq) on endometrial biopsies from live subjects or whole endometrium attached to underlying myometrium from deceased subjects who died of non-gynecological causes. They discovered similar clustering of genes as did Wang et al. [31], defining distinct cell populations that fall into epithelial, stromal, immune, supporting, and endothelial classes. They further parsed genetic changes that occur during the proliferative and secretory phase of the menstrual cycle by cell type, identifying various patterns, such as epithelial subtypes that preferentially express *SOX9* and *LGR5* in the proliferative phase, while representing other clusters of genes in the secretory phase to promote glandular secretions. Finally, Garcia-Alonso et al. determined transcriptional changes in individual cell types in an endometrial organoid model exposed to estrogen and progesterone and discovered key differences in transcription factor responses among different cell types. This study demonstrates the power of such organoid models in enhancing our understanding of tissue physiology with single-cell precision, and this is shown schematically in Figure 1B.

## 4. Endometrial Transcriptome in Endometriosis

Evidence has now started to accumulate supporting the role of the endometrial transcriptome in endometriosis development, maintenance, and progression. Extensive studies have attempted to examine the genetic makeup of the eutopic endometrium in women with endometriosis and explore its unique expression patterns compared with those of unaffected women. Using global messenger RNA (mRNA) sequencing, Zhao et al. pioneered a genome-wide gene expression profiling of the eutopic endometrium of women with endometriosis and discovered 72 differentially expressed genes thought to be implicated in core disease processes [34]. For instance, overexpression of genes involved in stromal invasion (matrix metalloproteinase *(MMP)-11*), estrogen-mediated endometrial proliferation (*FOS*), dysregulated fibrinolysis (*SERPINE1*), and angiogenesis (vascular endothelial growth factor *(VEGF) A*) were noted in their study, with the former three potentially representing novel biomarkers of the disease [34]. 

Likewise, a systematic review by May et al. compiled the results of 182 studies elucidating the endometrial differences in women with endometriosis and evaluating their candidacy as putative disease biomarkers [35]. The expression of various molecules has been shown to either increase or decrease in endometriosis, including cytokines, growth factors, angiogenic factors, reactive oxygen species, and many other mediators involved in apoptosis, tissue remodeling, and endometrial hormone regulation. For example, interleukin (IL)-8 and its receptor, which are engaged in chemoattraction, lesion growth, and angiogenesis, were determined to be overexpressed in endometriosis [36,37]. Likewise, elevated levels of miRNA of the chemokine Regulated upon Activation, Normal T-cell Expressed and Secreted (RANTES) were present in the secretory endometrium of women with endometriosis [38]. Several genes involved in mitosis and proliferation, and typically downregulated in the secretory endometrium of unaffected women, are upregulated in that of women with endometriosis, indicating a preferential tendency for endometrial cells in affected women to replicate even outside the proliferative phase [39].

Progesterone resistance is increasingly recognized as an inherent feature of endometriosis, and this correlation with disease pathogenesis has been evidenced to exist on genetic grounds [36]. Under physiological conditions, progesterone mediates direct inhibition of estrogen-induced DNA synthesis through downregulation of genes linked to DNA replication, primarily those from the minichromosome maintenance (*MCM*) family [39]. Burney et al., in their global gene expression analysis, showed an upregulation of all genes in that family in the early secretory endometrium in women with endometriosis, suggesting the persistence of proliferative characteristics in the endometrium of women with endometriosis that would typically transition to a differentiated state in an otherwise progesterone responsive state [39].

While most studies have examined the transcriptome of the eutopic endometrium in women with endometriosis, Ahn et al. studied the molecular profile of the ectopic endometrium, i.e., endometriotic lesions, from an inflammation and immune dysregulation perspective [40]. They concluded that genes encoding for several proinflammatory cytokines and receptors were significantly overexpressed in the ectopic lesions compared with both matched eutopic endometrium and control endometrial tissue. Similarly, genes closely linked to leukocytes and antigen-presenting cells, such as CD4, CD45R0, CD8A, CD40, and HLA, were overexpressed in ectopic lesions, which is consistent with the marked infiltration of these lesion by inflammatory cells [36,40].

Conversely, multiple studies have failed to identify consistent differences in the genetic makeup and gene expression of the endometrium of women with endometriosis compared to women without it [41,42]. In addition, if discovered, these differences may be cycle-dependent rather than disease-specific [43]. Fassbender et al., for instance, performed a combined mRNA and proteomic analysis of the eutopic endometrium in women with and without endometriosis and concluded that mRNA expression is comparable between the two groups; however, it was differentially expressed for 925 genes in women with endometriosis and for 1087 genes in women without the disease during menstruation compared with the early secretory phase [43]. Nevertheless, the authors acknowledge that including women with other gynecologic disorders, such as uterine fibroids, ovarian cysts, and hydrosalpinges, in their control group may have contributed to the lack of significant differences between cases and controls. From another clinically important perspective, their analysis guided in stratifying disease severity into minimal–severe, minimal–mild, and moderate–severe endometriosis utilizing five selected peptide peaks [43].

## 5. Endometrial Transcriptome in Recurrent Implantation Failure

Notable changes in the endometrial transcriptome have been demonstrated in patients with recurrent implantation failure, and those have been largely studied in association with endometriosis. Recipients of oocyte donation with recurrent implantation failure have been shown to have an intrinsically defective endometrial gene expression impeding successful implantation [44]. Some of these abnormally expressed genes include those encoding for C4b-binding protein (*C4BP*), glycodelin or progestogen-associated endometrial protein (*PAEP*), *MMP7*, and CXC chemokine receptor-4 (*CXCR4*) [44]. For example, in women who achieve spontaneous pregnancy, C4BP may exhibit inhibitory functions on the classical complement pathway, conferring protection against complement-mediated embryo attack [44]. This protein has in fact been shown to be downregulated in the receptive phase of women with endometriosis, possibly contributing to their infertility [45]. 

Glycodelin has been shown to exhibit embryo protective functions through immunosuppression and inhibition of NK cell activity [44,46], and its diminished endometrial expression may hence result in a heightened state of immunosurveillance that is determinantal to embryo viability. Endometrial CXCR4, on the other hand, possesses lymphocyte chemotactic properties and facilitates blastocyst adhesion during human implantation, and its expression levels in the human endometrium may therefore dictate trophoblast invasion and maintenance of pregnancy [44,47]. Kao et al. aimed in their study to reveal candidate genes implicated in endometriosis-associated implantation failure and infertility and were able to identify three gene clusters. The first cluster included genes normally upregulated during the implantation window. However, it is downregulated in women with endometriosis (e.g., IL-15, glycodelin, proline-rich protein, B61, and N-acetylglucosamine-6-O-sulfotransferase (*GlcNAc6ST*)), whereas the second cluster comprised genes that are conversely downregulated during the normal implantation window but upregulated in endometriosis (e.g., semaphorin E, neuronal olfactomedin-related endoplasmic reticulum localized protein mRNA, and Sam68-like phosphotyrosine protein ⍺). Lastly, the third group included a single gene, neuronal pentraxin II, commonly downregulated during the implantation window and further so in endometriosis [45]. The constellation of these differential gene expression in the eutopic endometrium of women with endometriosis render it an inhospitable milieu for embryo survival through mechanisms that are not restricted to direct embryo toxicity, immune dysregulation, exaggerated inflammatory responses, and enhanced apoptosis. In addition, GlcNAc6ST and olfactomedin-related protein may impair the tethering and attachment of the embryo as it adheres to the endometrium [45].

Elucidating maternal–fetal communication at the level of the decidua in early pregnancy will likely prove instrumental in understanding how normal implantation occurs and why some women experience recurrent implantation failure. Using scRNA-seq, Vento-Tormo et al. [48] first identified the different cell types at the maternal–fetal interface from five human placentas at 6–14 weeks of gestation. They identified perivascular cells located in spiral arteries as well as immune cells including CD8+ T lymphocytes and three subsets of decidual NK cells which likely orchestrate proper implantation and placentation via multiple cytokine and chemokine expression and secretion. The investigators identified three distinct subpopulations of dNK cells by flow cytometry that likely contribute uniquely to proper implantation. For instance, with the dNK1 subset secreting colony-stimulating factor-1 (CSF1), a factor whose expression correlates positively with implantation success [49]. 

Decidualization of the endometrium, a state of cell cycle arrest and cellular senescence, is an important physiological process in the menstrual cycle, but when dysregulated, it can lead to implantation failure. Lucas et al. [50] used single-cell transcriptomics to first map the effects of progestin-induced decidualization on endometrial stromal cells. They tracked changes in gene expression among these cells as the cells became senescent, finding genes important in energy regulation, ECM remodeling, and iron storage differentially expressed during decidualization. The investigators then compared endometrial biopsy samples of subjects with recurrent pregnancy loss (RPL) to controls, and discovered lower *SCARA5* and higher *DIO2* mRNA levels in the RPL group, consistent with a pro-senescent decidual phenotype. These genes may thus serve as pathogenic markers of implantation failure risk, and may also provide mechanistic insight into the pathogenesis of the condition. Along this line of investigation, Lucas’s group has recently developed an “endometrial assembloid” model, layering endometrial gland-like organoids and stromal cells to create an endometrial organoid system [51]. With this system, these investigators hormonally simulated a midsecretory endometrium (at a time in the menstrual cycle when implantation is most favorable) and sequenced glandular epithelial cells (EpCs) from this phase. They determined, based on transcriptome profiles, that five EpC subsets could be discerned, with dynamic gene regulatory changes corresponding to phase of the menstrual cycle. Acute cellular senescence led to production of the senescence-associated secretory phenotype (SASP), a milieu of immunomodulatory cytokines, chemokines, growth factors, and ECM proteins produced by EpCs, that ultimately allowed for invasion of the human blastocyst and successful implantation. Disruption of this cellular senescence programming thus likely contributes to implantation failure.

Transcription profiles have been compared within the endometrium from control subjects to those with recurrent implantation failure, but also to the peripheral blood transcriptome. Huang et al. [52] reported that in controls and those with recurrent implantation failure, transcriptome profiles of peripheral blood correlated with endometrium for about 10% of genes. While endometrium profiles were distinct between controls and those with recurrent miscarriages or recurrent implantation failure, peripheral blood samples were harder to distinguish between these groups.

Notably, the evidence in this avenue has not always come in support of the role of altered endometrial transcriptome in endometriosis-associated infertility. Through mRNA and miRNA sequencing of 17 endometrial samples, Da Broi et al. concluded that the eutopic endometrium of infertile women with endometriosis may not be transcriptionally discrepant from the endometrium of both fertile and infertile controls during the implantation window [53]. While this may introduce contradictory conclusions to the existing literature, the authors acknowledge that this data discrepancy may be attributed to several reasons, including large sample heterogeneity between studies, cycle-dependent variations, and differences in selection criteria wherein controls may have other endometrial transcriptome-altering conditions that potentially mask existing significant differences [53]. 

## 6. Interplay between Endometrium and Immune System Revealed through Transcriptomics

As alluded to above, the immune system fluctuates in synchrony with the menstrual cycle in subtle yet significant ways that hint at an important role for immune cells in the physiological process of menstruation and pregnancy establishment. During the proliferative phase of the menstrual cycle, circulating regulatory T cells (Tregs) are relatively more abundant along with increased plasma IL-1 levels compared to the secretory phase [54,55]. The significance of these cells has been suggested by the association between recurrent pregnancy loss and low circulating Tregs, likely representing a breakdown of semi-allogeneic tolerance. However, a causal link has yet to be definitively drawn [55,56]. Furthermore, in a subset of patients with polycystic ovary syndrome (PCOS), which is typified by menstrual irregularities such as oligo-ovulation or anovulation, an imbalance of T_H_17 and Treg cells has been proposed [57].

The complex interplay between the immune system and menstruation has been elucidated in recent years with the implementation of RNA sequencing. As the interpretation of RNA profiles depend greatly on relative expression of genes or differentially expressed genes (DEGs), it becomes important to define a baseline gene expression profile and compare it to an experimental condition or disease state. In the field of female reproductive and immunological physiology, endometriosis has become perhaps the most extensively studied disease state with regard to RNA transcriptome analysis.

To study immunological differences in menstruation between healthy subjects and those with endometriosis, Miller et al. [58] recruited healthy volunteers without endometriosis and those with endometriosis and collected menstrual effluent (ME) from these subjects during their menstrual periods. ME was centrifuged, with ME cells separated from supernatant “serum”, and multiplex cytokine analysis was conducted. These investigators uncovered a wide variety of cytokines present in the samples. While the expression of most cytokines was similar between groups, TGF-alpha was significantly lower in ME serum of patients with endometriosis compared to that of controls. Under physiological conditions, TGF-alpha concentrates on the surface epithelium of the endometrium during the proliferative phase and declines during the secretory phase [59]. Thus, altered levels of TGF-alpha on the surface endometrium may correspond to reduced endometrial receptivity in cases of endometriosis, potentially contributing to reduced fertility. 

The authors also used flow cytometry to identify T_H_17 cell populations from the ME samples and compare their transcriptome profiles. They discovered no change in the overall number of T_H_ cells between groups but did report that the T_H_17 subpopulation was significantly lower in the ME of subjects with endometriosis compared to those of patients without endometriosis. They further identified 47 DEGs in the MEs between groups, and in particular determined that genes related to the T_H_17 axis such as *IL10*, *IL23A*, and *IL6* were downregulated in subjects with endometriosis. Likewise, macrophage-associated genes such as *CD74*, *CD83*, *CXCL16* and *CCL3* were downregulated in the ME of subjects with endometriosis [58].

Endometriosis is characterized not only by changes in the eutopic endometrium lining the inner surface of the uterus, but also by foci of endometrial glands and stroma that can be distributed throughout the abdominal cavity. To examine how both eutopic and ectopic endometriosis differ from healthy endometrium in cellular components and transcriptomics, Tan et al. collected human control endometrium as well as eutopic and ectopic endometrium from surgical specimens taken from subjects with endometriosis. They performed tissue dissociation for single-cell RNA-seq as well as bulk RNA-seq analyses on tissue samples. They determined that the transcriptome profile of endometrial cells differs between eutopic endometrium of patients with endometriosis (EuE) and controls without endometriosis, specifically with increased expression of cell-cycle-related genes and proliferation of endometrial fibroblasts in the EuE group. Osteoglycin expression was determined to be higher in EuE versus control. Genes regulating immune cell attachment and monocyte trafficking and those associated with endothelial cell permeability were also upregulated in endometriosis. Finally, *AQP1*, involved in angiogenesis and endothelial cell migration, was determined to be significantly upregulated in endometriosis. 

Their transcriptome analysis identified five different subtypes of macrophages, with macrophage 1-LYVE1 upregulating tolerogenic genes and angiogenesis-promoting genes. Dendritic cells adjacent to ectopic endometrium adopt an immunomodulatory phenotype, upregulating phagocytosis and cytokine signaling pathway-related genes. The authors further characterized a unique subpopulation of MUC5B-expressing endometrial epithelial cells and implicated these cells in epithelial repair immune cell recruitment via *TFF3* (trefoil factor 3) and *SAA* (serum amyloid A) [60].

A final notable cell type that plays a role in normal menstrual physiology and immune tolerance is the decidual natural killer (dNK) cell. Maturation and proliferation of dNK cells during the first trimester of pregnancy allows not only for spiral artery growth to feed the developing fetus, but also protection against the maternal host immune response [61,62]. On the other hand, abnormally activated NK cells can lead to miscarriage and fetal alloimmune thrombocytopenia by inducing trophoblast apoptosis [62]. Single-cell transcriptome analysis has further clarified that dNK cellular mechanisms dictate uterine spiral artery organization and pregnancy fate. Pan [63] collected control and unexplained recurrent spontaneous abortion (URSA) decidua samples, performed tissue dissociation and single-cell isolation, then carried out molecular barcoding and sequencing of samples. Using a bioinformatics analysis, they identified 29 distinguishable clusters of genes from dNK cells. Not only were the proportions of total and proliferating dNK cells higher in the URSA group compared to control, but they also identified DEGs in dNK cells of URSA samples including *C1orf162*, *RASGEF1B*, and *ZNF683* suggestive of abnormal differentiation, and lower expression of *COL1A1*, *COL3A1*, and *PAEP*, suggestive of abnormal extracellular matrix and spiral artery reorganization. Together, these findings point to changes in dNK cell programming that may underlie recurrent pregnancy loss. This single-cell RNA sequencing approach could also serve as a tool to help predict groups at risk for pregnancy loss and to target specific genes for dNK cell reprogramming for future therapeutic potential [63]. Characteristic transcriptional changes seen in endometriosis related largely to immune function are shown schematically in Figure 2.

## 7. Transcriptome Analysis in Other Benign Gynecological Conditions

### 7.1. Fibroid Transcriptomics

Uterine fibroids are common benign monoclonal tumors composed of smooth muscle cells in the myometrium [64]. Many women with uterine fibroids do not exhibit any symptoms [65]. Those who are symptomatic can experience abnormal bleeding, pain, urinary symptoms, and sometimes unfavorable pregnancy outcomes such as infertility and pregnancy loss [66,67,68,69]. Uterine fibroids can be described as an endometrial cavity distorting leiomyoma (ECDL) or endometrial cavity non-distorting leiomyoma (ECNDL); these categorizations indicate whether the fibroid is affecting the endometrial cavity or related to differences in infertility [70]. 

A recent transcriptome-wide association study (TWAS) conducted by Kim et al. [71] including roughly 20,000 patients with fibroids and 223,000 control subjects identified nine fibroid susceptibility genes, including two novel genes, *RP11-282O18.3* (a long non-coding RNA that is involved in immune and hormonal regulation via estradiol) and *KBTBD7* (encoding a transcriptional activator that participates in Ras/ECM signaling to mediate tumorogenesis). These investigators also discovered changes in expression patterns of several microRNAs (miRNAs), the relevance of which we now discuss in greater detail.

There is growing evidence to support the idea that miRNAs, non-coding RNA molecules that play a significant role in intracellular signaling, apoptosis, growth and metabolism, are involved in uterine fibroid pathogenesis and pregnancy-related complications [72,73,74,75,76]. Uterine fibroids have a distinct miRNA expression compared to normal myometrium. There is an upregulation of miR-15b and downregulation of *miR-29a*, *-29b*, *-29c*, *197*, and *-200c* in uterine fibroids compared to the normal myometrium [77]. These miRNAs are related to cell proliferation and division, apoptosis, migration, invasion, metabolism, stress, angiogenesis, and drug resistance [77]. Additionally, mi-R-29 is related to the extracellular matrix (ECM) remodeling, and miR-197 and -200 are related to cell proliferation and suppression of tumor development [77,78,79]. Certain miRNAs have further been shown to be upregulated in TGFBR2+ and IGF2BP1+ fibroids, supporting the hypothesis that miRNAs are important in TGFB-SMAD and growth factor signaling for cellular growth and proliferation of fibroids [74]. MiRNA also plays a role in the gene expression profiles of estrogen and progesterone receptors and regulators of ECM as these genes are upregulated in uterine fibroids compared to normal myometrium [77]. This finding suggests the importance of miRNA in modifying the fibroid hormonal micro-environment and ECM composition [77]. Together, the transcriptome data indicate the critical role miRNA expression plays in uterine fibroid development.

MiRNA expression profiles differ in ECDL and ECNDL cells [73]. Both cell populations display the downregulation of miR-29b and -200c expression compared to normal myometrial cells, but ECDL cells show more downregulation of these miRNAs compared to ECNDL (Kim et al., 2018). ECDL cells also show an upregulation of *E-Rcβ*, *MMP1*, and *TIMP-2* expression—genes deemed important in uterine fibroid development [77,80,81,82]. These data suggest miR-29b and -200c may control *E-Rcβ, MMP1*, and *TIMP-2* expression [77]. ECDLs may affect endometrial receptivity by impacting the uterine structure and leading to decreased embryo implantation and pregnancy, though further miRNA expression studies of this cell population should be performed [77]. On the other hand, the endometrial transcriptome of women with ECNDLs reveals no differences in endometrial receptivity or in genes examined through ERA in ECNDL cells compared to normal endometrial cells [83]. Additionally, proliferation and apoptosis rates, decidualization markers, and the morphological transformation of endometrial stromal fibroblasts (eSF) in response to decidualization stimuli are not different in ECNDL tissue compared to normal endometrial tissue [83]. Therefore, ECNDL uterine fibroids do not negatively impact the endometrial transcriptome and hence uterine receptivity and implantation [83]. Perhaps the miRNA expression differences between ECDL and ECNDL cells may explain some of the symptomatic differences seen clinically in women who have ECDL versus ECDNL fibroids. 

MiRNA expression is not the only aberrant RNA change affecting the female reproductive tract in patients with fibroids. Changes in the endometrium itself have been observed in patients with fibroids. For instance, Aghajanova et al. [83] determined that ECNDL are associated with over 4000 differentially regulated genes in the endometrium compared to control endometrium of non-fibroid containing uteri. Of these, notably upregulated genes include *JUN* and *FOS*, important in cell cycle progression, as well as genes involved in cell differentiation, cell survival, and angiogenesis. By comparison, for cavity-distorting fibroids (e.g., submucosal), endometrial *HOXA10* and *HOXA11* mRNA levels have been discovered to be significantly decreased compared to control uteri, suggesting a global transcription factor-dependent mechanism by which fibroids may lead to infertility given that these genes are involved in trophoblast invasion [84]. 

### 7.2. Adenomyosis Transcriptomics

Adenomyosis is a disease caused by the infiltration of endometrial tissue into myometrial tissue. It can cause various symptoms, including abnormal uterine bleeding, pelvic pain, and infertility in female individuals of reproductive age [85,86]; or can be asymptomatic. The current standard treatment of adenomyosis is hysterectomy, and there are no established medical therapies to treat adenomyosis [85]. 

Adenomyosis is present in 24.4% of female individuals with infertility, leading many to use in vitro fertilization (IVF) to aid in their efforts to become pregnant [87,88]. It is believed that adenomyosis is related to decreased IVF outcomes, but this relationship is not entirely understood because many existing studies did not control for factors such as age, degree of adenomyosis, coexisting pelvic disorders, number of IVF cycles conducted, et cetera [89]. Before an IVF cycle is started, an endometrial receptivity array (ERA) can be performed to elucidate the transcriptomic expression of 238 genes related to endometrial receptivity status. The receptivity status will determine whether a patient receives standard embryo transfer or personalized embryo transfer (PET) [90]. A study performed by Juarez-Barber and colleagues sought to understand if women with adenomyosis experienced changes in endometrial receptivity and if PET by progesterone adjustment based on transcriptomic analysis improved IVF outcomes. The study demonstrated that in patients with adenomyosis, there are significant changes in endometrial receptivity, increasing the risk of developing non-receptive endometria [15,91]. However, the use of transcriptome-based progesterone administration timing before PET does not improve IVF outcomes (implantation, biochemical and clinical miscarriage, or live birth weights) in women with adenomyosis [15]. Alterations in endometrial receptivity-related gene profiles have been determined to be largely non-overlapping compared to those seen in endometriosis, with several key exceptions including *SCGB2A2* and *NR4A1* [92]. These results suggest other hormones or pathways may be responsible for implantation failures detected in patients with adenomyosis [15]. Another study employing single-cell RNA-seq of eutopic endometrial samples from patients with adenomyosis identified clusters of epithelial and endothelial cells with differentially expressed genes compared to ectopic endometrium from the myometrial tissue, with the latter enriched for genes important for angiogenesis and cell growth, suggesting that myometrial and eutopic endometria may exhibit distinct physiological responses to hormonal changes, with ectopic endometrium behaving in a less tightly regulated fashion [93].

## 8. Conclusions

In this article, we have reviewed basic reproductive physiology and common gynecological disorders through the lens of transcriptome analysis. The strength of this approach is the ability to build maps of complex, interlocking signaling networks through relative expression of genes within the endometrium. As the endometrium is a dynamic tissue that ebbs and flows with hormonal fluctuations during the menstrual cycle, it is unsurprising that such plasticity would be accompanied by significant shifts in gene expression in pathways related to cell cycling, leukocyte recruitment, and receptivity in anticipation of a possible implanting blastocyst. Understanding the natural transcriptome cycle as it relates to the menstrual cycle has aided our understanding of disease states such as endometriosis, adenomyosis, and uterine fibroids. In such cases, inflammatory cascades, abnormal immune cell recruitment, and disruption of normal endometrial remodeling set the stage for symptomatology such as infertility, pelvic pain, and abnormal uterine bleeding. As precision medicine rises to the forefront, transcriptome profiling will likely allow us to more completely understand ways in which changes in gene expression on a large scale give rise to a spectrum of gynecological disorders that differs from one patient to another.

## Figures and Tables

**Figure 1 biology-12-00634-f001:**
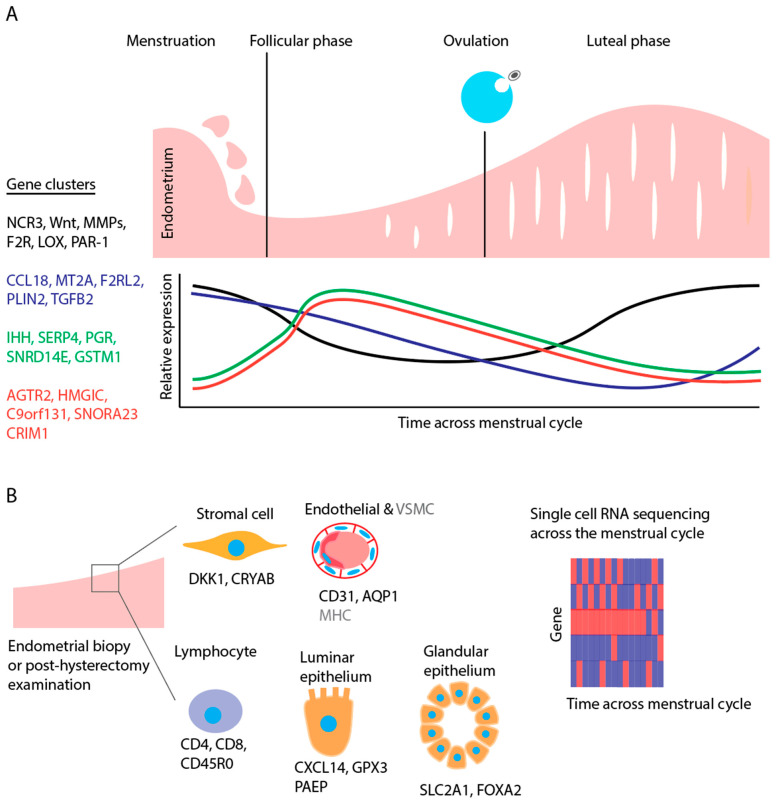
Changes in gene expression profiles in the endometrium across the menstrual cycle. (**A**) During the secretory phase and during menstruation, genes related to apoptosis, inflammation, signal transduction, transcription, DNA repair, and extracellular matrix remodeling are upregulated. During the early proliferative phase, genes needed for the regeneration of the endometrial functional layer after menstruation are upregulated. Gene expression involved in cell renewal processes such as cell proliferation, cell survival, and regulation of differentiation are upregulated in the mid-proliferative phase. Finally, in the late proliferative phase, genes linked with inhibition of the cell growth, extracellular matrix remodeling, and cellular differentiation are upregulated. (**B**) Single-cell RNA sequencing (scRNA-seq) of endometrial tissue components across the menstrual cycle revealed characteristic and cell type-specific changes in gene expression related to the proliferative and secretory phases of the cycle. VSMC, vascular smooth muscle cell. Heat map: red represents relatively increased gene expression and blue represents relatively decreased gene expression. The X axis is relative time across the menstrual cycle, and the Y axis is discrete genes.

**Figure 2 biology-12-00634-f002:**
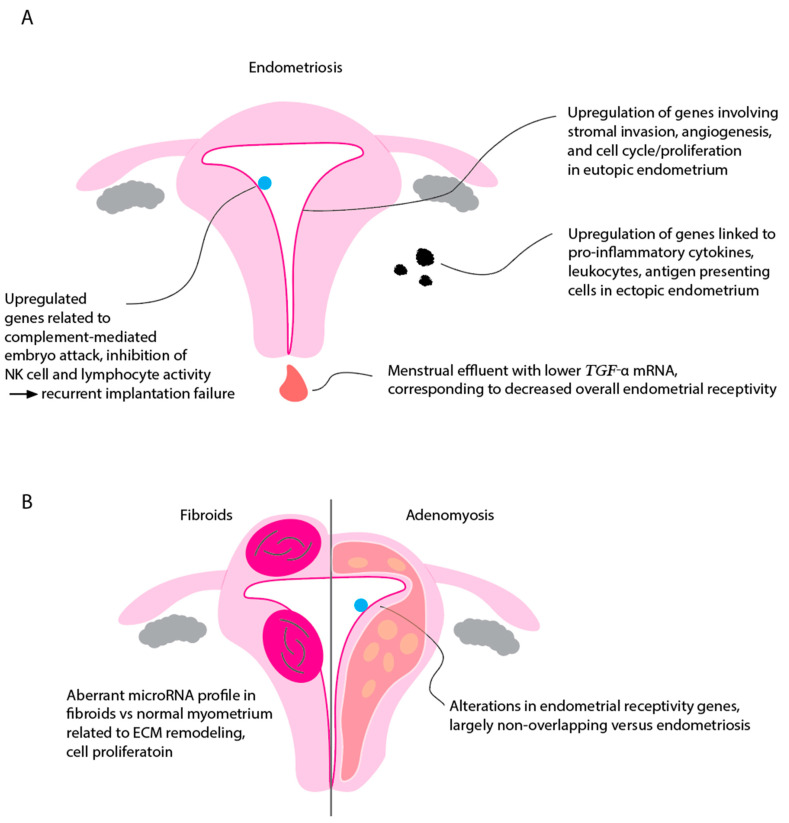
Transcriptomic changes in common benign gynecological disorders. (**A**) Alterations in gene expression in eutopic endometrium, ectopic endometrial implants (dark “powder burn” lesions for example), and menstrual effluent are displayed. (**B**) Changes in microRNA profiles in uterine fibroids, and changes in genes related to endometrial receptivity in adenomyosis are displayed.

## Data Availability

Not applicable, no original data are provided.

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
