# Peer review of "Uterine Transcriptome: Understanding Physiology and Disease Processes"

_biology, 2023, doi:10.3390/biology12040634_

Round 1
Reviewer 1 Report
In the review, the authors reviewed literature describing endometrium in both normal and disease conditions. The review summarizes the role and physiology of the tissue, as well as the methods used to measure gene expression in the endometrium in the context of endometriosis and recurrent implantation failure. The authors reviewed transcriptional changes observed in benign gynecological conditions, such as fibrosis and adenomyosis.
Overall, the review is well-written and reads smoothly. However, I have a few suggestions for the authors to consider:
1. In figure 1, the authors describe gene expression profiling in the endometrium across the menstrual cycle. This information is valuable for scientists analyzing transcriptomic data. Could the authors refer more explicitly to data sets that could be used to identify or classify gene expression data in relation to the menstrual cycle?
2. Could the authors comment on the possible correlation in gene expression between the endometrium, menstrual effluent, and peripheral blood? What is the purpose of using these sample types, and how informative are they?
3. In Figure 1B, it is unclear to non-specialists in single-cell sequencing what the heatmap represents below the legend "Single-cell RNA sequencing across the menstrual cycle." Could the authors provide more clarity on this point?
4. Could the authors give more information about why most studies on fibrosis have focused on miRNA sequencing, and whether gene expression analysis supports the results published on miRNA expression??
Author Response
April 16, 2023
Associate Editor
Biology
RE: Manuscript ID: biology-2250097
Entitled; Endometrial Transcriptome: Understanding Physiology and Disease
Processes
Dear Associate Editor,
First, we would like to thank the Associate Editor and the Reviewers for their helpful and constructive comments. We have revised the manuscript to incorporate the valuable comments from the Editor and Reviewers. We believe that manuscript now is more robust and hope it is now ready for publication.
Below is a line-for-line response to comments from the editor and reviewers:
In the review, the authors reviewed literature describing endometrium in both normal and disease conditions. The review summarizes the role and physiology of the tissue, as well as the methods used to measure gene expression in the endometrium in the context of endometriosis and recurrent implantation failure. The authors reviewed transcriptional changes observed in benign gynecological conditions, such as fibrosis and adenomyosis.
Overall, the review is well-written and reads smoothly. However, I have a few suggestions for the authors to consider:
- In figure 1, the authors describe gene expression profiling in the endometrium across the menstrual cycle. This information is valuable for scientists analyzing transcriptomic data. Could the authors refer more explicitly to data sets that could be used to identify or classify gene expression data in relation to the menstrual cycle?
We have added a line in the text (pages 7-8) that references work from Wang et al. 2020 (Nature Medicine) that can be used to analyze such transcriptomic data. See NCBI’s Gene Expression Omnibus seris accession code GSE111976.
- Could the authors comment on the possible correlation in gene expression between the endometrium, menstrual effluent, and peripheral blood? What is the purpose of using these sample types, and how informative are they?
The reviewer raises an interesting question regarding how gene expression profiles may differ or be correlated between peripheral blood, menstrual effluent, and peripheral blood. We have added a references to address this. Huang et al. 2017 (PLOS ONE) showed that in women with recurrent miscarriages and recurrent unexplained implantation failure (compared to controls), transcriptome profiles of peripheral blood correlated with endometrium for about 10% of genes, and while endometrium profiles were distinct between controls and those with recurrent miscarriages or recurrent implantation failure, peripheral blood samples were fairly indistinguishable between the three groups. We have added this information to the text on page 13.
- In Figure 1B, it is unclear to non-specialists in single-cell sequencing what the heatmap represents below the legend "Single-cell RNA sequencing across the menstrual cycle." Could the authors provide more clarity on this point?
We thank the reviewer for pointing out the lack of clarity. We have added a key to Fig 1B explaining that red represents relatively increased gene expression and blue represents relatively decreased gene expression (page 27).
- Could the authors give more information about why most studies on fibroids have focused on miRNA sequencing, and whether gene expression analysis supports the results published on miRNA expression??
We have included rationale for why extensive investigation has been performed on miRNA sequencing in uterine fibroids. Dysregulation of miRNAs in fibroids has been proposed to partially explain the pathogenesis of the disease. In particular, several miRNAs (miR-181a-5p, 127-3p, 28-3p, 30b-5p and let-7c-5p) are upregulated in TGFBR2+ and IGF2BP1+ fibroids, lending credence to the idea that miRNAs are important in TGFB-SMAD and growth factor signaling for cellular growth and proliferation (Kim et al., 2022 PLOS ONE). With regard to the reviewer’s second point, there has been a recent transcriptome-wide association study of uterine fibroids conducted by Kim et al. (2022, POS ONE), which identified fibroid susceptibility genes, including two novel genes, RP11-282O18.3 and KBTBD7, which may be implicated in fibroid pathogenesis. We have included these references in our revised manuscript and make these points on page 17.
Reviewer 2 Report
Kirschen et al. provide a helpful literature review of studies of the human endometrium that have utilized transcriptomic approaches. After brief introductions to endometrial biology and transcriptomic techniques, the review covers studies on changes in the normal endometrial transcriptome through the menstrual cycle before turning to transcriptomic analysis of gynecological disease states. Discussion of a good sampling of the most important and relevant studies is included, and the review helps to synthesize the results for easier comprehension. However, the review is not comprehensive for recent studies on the topics it addresses, and it is not clear how the authors chose the articles included in the review.
General comments:
1. No search methods are reported. The authors should describe how they found and selected the primary literature that was included in this review.
2. The discussions of recurrent implantation failure and the interplay between the endometrium and immune system are missing three key recent studies which have helped shed light on the function of natural killer cells in early pregnancy (PMID: 30429548, 31965050, 34487490. Including a discussion of these articles would benefit the review and make it more comprehensive.
3. Some of the content under the subheading “4.1 Fibroid transcriptomics” focuses on fibroid development and myometrial biology rather than the endometrium. If the authors wish to focus on the endometrium, this section should be edited to limit the discussion to the effects of fibroids on the endometrium. If they want to keep the broad discussion, perhaps they would consider adjusting the article title to reflect the inclusion of studies outside the endometrium.
Specific comments:
1. The statement is made that decidualization is “cyclic AMP (cAMP)-dependent” (line 43); however, recent studies show that PGE2 can replace it in vitro and argue that it is more physiological (PMID: 34591094, 36821428). These studies utilize transcriptomic approaches, and discussing them would add value to this article.
2. The citation of article “[9]” on line 52 appears to be in error. The cited reference does not directly cover the topic of the preceding sentence.
3. The statement, “therefore, they may be implicated in endometrial gland development” seems redundant in the context of the earlier part of the sentence on lines 112-114.
4. The information value of Figure 1B is low in its current form. It would be very useful if the authors would summarize the single cell RNA-sequencing findings discussed in the text in figure form by identifying the most common genes that are associated with each cell type shown.
5. The terms “proliferative” and “secretory” are most frequently used for the phases of the menstrual cycle, but “follicular” and “luteal” are also sometimes used. Since this article focuses on the endometrium and not the ovary, using “proliferative” and “secretory” in all cases would help with continuity.
Author Response
Kirschen et al. provide a helpful literature review of studies of the human endometrium that have utilized transcriptomic approaches. After brief introductions to endometrial biology and transcriptomic techniques, the review covers studies on changes in the normal endometrial transcriptome through the menstrual cycle before turning to transcriptomic analysis of gynecological disease states. Discussion of a good sampling of the most important and relevant studies is included, and the review helps to synthesize the results for easier comprehension. However, the review is not comprehensive for recent studies on the topics it addresses, and it is not clear how the authors chose the articles included in the review.
General comments:
- No search methods are reported. The authors should describe how they found and selected the primary literature that was included in this review.
We thank the reviewer for their comment. We have added a statement in our Background regarding how the literature review was conducted. Briefly, we performed an extensive search of the PubMed MEDLINE database using relevant search terms that included “transcriptome”, “transcriptomics”, “RNA”, “RNA sequencing” “microRNA”, “menstrual cycle”, “endometriosis”, “leiomyomas”, “fibroids”, “adenomyosis”, “recurrent implantation failure”, and “gynecological”. We also note that additional relevant articles have been added during the revision process as suggested by the expert reviewers. See page 5.
- The discussions of recurrent implantation failure and the interplay between the endometrium and immune system are missing three key recent studies which have helped shed light on the function of natural killer cells in early pregnancy (PMID: 30429548, 31965050, 34487490. Including a discussion of these articles would benefit the review and make it more comprehensive.
We thank the reviewer for pointing out these relevant and critical studies. In our section on recurrent implantation failure, we have extensively expanded upon this discussion, incorporating these studies. See pages 12-13.
- Some of the content under the subheading “4.1 Fibroid transcriptomics” focuses on fibroid development and myometrial biology rather than the endometrium. If the authors wish to focus on the endometrium, this section should be edited to limit the discussion to the effects of fibroids on the endometrium. If they want to keep the broad discussion, perhaps they would consider adjusting the article title to reflect the inclusion of studies outside the endometrium.
We thank the reviewer for this suggestion. We have broadened our title to “Uterine Transcriptome” to more accurately reflect the discussion of both endometrial and myometrial transcriptional changes that we discuss throughout the review.
Specific comments:
- The statement is made that decidualization is “cyclic AMP (cAMP)-dependent” (line 43); however, recent studies show that PGE2 can replace it in vitro and argue that it is more physiological (PMID: 34591094, 36821428). These studies utilize transcriptomic approaches and discussing them would add value to this article.
We have made the change, as suggested (page 4).
- The citation of article “[9]” on line 52 appears to be in error. The cited reference does not directly cover the topic of the preceding sentence.
We thank the reviewer for pointing out this error, which we have amended.
- The statement, “therefore, they may be implicated in endometrial gland development” seems redundant in the context of the earlier part of the sentence on lines 112-114.
We have removed this phrase from the text.
- The information value of Figure 1B is low in its current form. It would be very useful if the authors would summarize the single cell RNA-sequencing findings discussed in the text in figure form by identifying the most common genes that are associated with each cell type shown.
We agree with the reviewer, and we have modified Figure 1B accordingly to highlight common genes expressed by each cell type.
- The terms “proliferative” and “secretory” are most frequently used for the phases of the menstrual cycle, but “follicular” and “luteal” are also sometimes used. Since this article focuses on the endometrium and not the ovary, using “proliferative” and “secretory” in all cases would help with continuity.
We have made the suggested change for consistency throughout the text.